# Potential of Bioassays to Assess Consequences of Cultivation of *Acacia mangium* Trees on Nitrogen Bioavailability to *Eucalyptus* Trees: Two Case-Studies in Contrasting Tropical Soils

**DOI:** 10.3390/plants12040802

**Published:** 2023-02-10

**Authors:** Kittima Waithaisong, Agnès Robin, Louis Mareschal, Jean-Pierre Bouillet, Jean-Michel Harmand, Bruno Bordron, Jean-Paul Laclau, José Leonardo Moraes Gonçalves, Claude Plassard

**Affiliations:** 1Eco&Sols, Institut Agro, Univ Montpellier, CIRAD, INRAe, IRD, 34060 Montpellier, France; 2CIRAD, UMR Eco&Sols, F-34398, 34000 Montpellier, France; 3ESALQ, University São Paulo, Piracicaba 13418-900, SP, Brazil; 4CRDPI, Pointe Noire 1291, Congo; 5School of Agricultural Sciences, UNESP-São Paulo State University, Botucatu 18610-307, SP, Brazil; 6Ecologie Fonctionnelle et Biogéochimie des Sols, 2 Place Pierre Viala, CEDEX 01, 34060 Montpellier, France

**Keywords:** soil N-mineralization rate, soil respiration rate, N_2_-fixing tree species, plant N accumulation

## Abstract

We hypothesized that the nitrogen-fixing tree *Acacia mangium* could improve the growth and nitrogen nutrition of non-fixing tree species such as *Eucalyptus*. We measured the N-mineralization and respiration rates of soils sampled from plots covered with *Acacia*, *Eucalyptus* or native vegetation at two tropical sites (Itatinga in Brazil and Kissoko in the Congo) in the laboratory. We used a bioassay to assess N bioavailability to eucalypt seedlings grown with and without chemical fertilization for at least 6 months. At each site, *Eucalyptus* seedling growth and N bioavailability followed the same trends as the N-mineralization rates in soil samples. However, despite lower soil N-mineralization rates under *Acacia* in the Congo than in Brazil, *Eucalyptus* seedling growth and N bioavailability were much greater in the Congo, indicating that bioassays in pots are more accurate than N-mineralization rates when predicting the growth of eucalypt seedlings. Hence, in the Congo, planting *Acacia mangium* could be an attractive option to maintain the growth and N bioavailability of the non-fixing species *Eucalyptus* while decreasing chemical fertilization. Plant bioassays could help determine if the introduction of N_2_-fixing trees will improve the growth and mineral nutrition of non-fixing tree species in tropical planted forests.

## 1. Introduction

Forest plantations are rapidly expanding to meet the increased demand for wood products and contribute to limit the deforestation of primary forests [1,2]. Among the most cultivated forest species, eucalypt trees, which cover about 20 million hectares [3], are mainly established in highly weathered tropical soils, generally poor in nutrients (in particular N and P). As rotations progress, eucalypt plantations can drastically reduce soil N, P, and K levels [4]. Although plants have adapted to overcome nutritional limitations, with mechanisms for reabsorption, biological recycling and allocation, and the utilization of N, P, and K, this is not always sufficient for high productivity [5]. In recent decades, two main methods have been used to overcome the lack of nutrients: minimum tillage [4] and the addition of mineral fertilizers [5,6]. The main problems with fertilizers are the high rates required by plants and their increasing cost. As an alternative to fertilizer use, a third option has emerged through the association of eucalypts with nitrogen-fixing species (NFS) [7,8]. Indeed, if high enough, soil N inputs from N_2_ fixation could reduce the environmental cost of producing and applying mineral N fertilizer in the field. The contribution of mineral fertilization in eucalypt forest to global warming has been estimated to be about 20% of total emissions, from forest planting to log exporting [9]. Nitrogen fixation has also been proposed as a way to reduce greenhouse gas (GHG) emissions related to fertilizer manufacturing (1.6 to 6.4 kg of CO_2_ equivalent per kg of nitrogen manufactured by chemical reaction [10]), in addition to avoiding the release of nitrous oxide (N_2_O) resulting from the application of fertilizers that are not synchronized with plant requirements [11]. Taken together, these data suggest that introducing NFS species could enhance N availability for eucalypt plantations without increasing the environmental cost or pollution of plantations.

Today, the success of forest plantations still depends on the application of mineral fertilizers [12]. Adequate fertilization is necessary for the long-term production of eucalypt plantations [13]. Several studies have shown that N fertilization can increase growth in early *Eucalyptus* plantations [13,14]. However, in some cases, this response is not maintained until the end of the rotation due to other limiting factors, such as soil water availability in Brazil [7]. Eucalypt trees require a large amount of nutrients in their initial growth phase [15] due to large investment in leaf biomass and root development during this period [6,16]. This high initial growth is possible due to the ability of young trees to take up large amounts of nutrients released from the mineralization of organic matter residues left over from the previous rotation [16]. In the second part of the eucalypt plantation cycle (after the third year), the growth rate of trees decreases, mainly because leaf production reaches a plateau, thus decreasing N demand [17]. During this period, the nutrient demand is largely met by the biogeochemical cycling of elements in the soil and mineralization processes of organic matter on the one hand, and by the biochemical cycling and internal retranslocation of nutrients within the tree on the other. Eucalypt trees rapidly explore the soil layers [18,19,20], resulting in low nutrient leaching from forest plantations [16,21]. When studying the rates of fertilizer application in eucalypt plantations, da Silva et al. [13] found that trees responded positively to increasing doses of fertilizers, resulting in higher productivity; however, the effects of these increasing doses decreased in the second year after planting. The split application of fertilizers in the first two years after planting did not increase plantation productivity compared with a single application at planting.

The association of NFS and non-NFS can be a form of ecological intensification, a process that aims to increase sustainable forest plantation production and soil nutrient availability [7,22,23,24]. The intercropping of NFS in eucalypt plantations, e.g., by planting one tree of each species alternately in the same row to obtain a plantation with 50% of each species, can increase biomass production, soil mineral status, and soil carbon, while reducing fertilizer costs [25,26,27]. For example, the association of these species with eucalypts has been shown to result in increased N availability through N_2_ fixation [28], as well as increased litter production [27], increased leaf decomposition, and increased cycling [23], resulting in increased soil N mineralization [26,29].The goal of mixed-species plantings is therefore to combine certain species, locations and attributes (temperature, precipitation, and soil) to maximize the balance between positive and negative interspecific interactions to increase individual tree growth and stand production, as well as reducing insect impacts or illnesses, and thus increasing the chances of plantation success [24,27]. However, despite their effectiveness, less than 0.1% of forest plantations in the world are mixed plantations [30,31].

As already mentioned, growing Acacia trees over 10 years after several decades of eucalypt plantations greatly influenced soil N [32,33] and organic P [34,35] statuses in two experiments in Brazil and the Congo. A possible option for improving N availability for eucalypt in plantations could be to insert an *Acacia* rotation (lasting 5–8 years) between successive eucalypt rotations. However, it is necessary to demonstrate that N bioavailability to eucalypt trees is actually improved by *Acacia* rotation compared with fertilization practices commonly used by forest managers. An effective way of measuring actual N bioavailability is to grow the target species on soil samples of interest and measure N accumulation in the plants [36,37,38,39,40].

We put forward the hypothesis that planting *Acacia* trees after several decades of eucalypt cultivation increases soil N bioavailability to eucalypt trees. To check this hypothesis, we first measured soil N-mineralization rates under laboratory conditions in soil samples collected from plots covered with *Acacia*, *Eucalyptus*, or native vegetation for at least 10 years. We also measured soil respiration as an indicator of microbial activity. Then, we assessed the actual N bioavailability of these soils for eucalypt seedlings grown in pots for at least 6 months and compared the results with those obtained with nonlimiting N fertilization to assess how soil N supply limited *Eucalyptus* growth in these soils.

## 2. Results

### 2.1. Soil N Mineralization Rates and Soil Respiration Rates

Total N (NH_4_^+^ + NO_3_^−^)-mineralization rates (Figure 1A) did not vary by vegetation type in Brazil and were significantly higher than those measured in the Congo in *Acacia* (Ac) and *Eucalyptus* (Euc) soils. Under native (Nat) vegetation, average total N-mineralization rates were twice as high in Brazil as the Congo. However, due the high variability of values measured in soil collected in Itatinga, no significant difference was found between the two sites. In contrast to Itatinga, total N-mineralization rates at Kissoko depended upon the vegetation type in the order Ac > Euc > Nat. As with total N, NH_4_^+^ release rates (Figure 1B) did not vary by vegetation at Itatinga, accounting for 16% (Ac soil) and 32–35% (Euc and Nat soils) of total N, indicating that nitrification was dominant in these soils. At Kissoko, NH_4_^+^ release rates varied significantly by vegetation type. The rates in Euc and Ac soil were the highest and the lowest, respectively, and those in Nat soil did not differ significantly from Ac and Euc soils. Regarding the proportions of NH_4_^+^ to total N release, they ranged from 20% in Ac soils to 66% in Euc soils, increasing up to 80% in Nat soils. This indicates that nitrification was not significant in Nat soils in the Congo.

Soil respiration rates were consistently higher in Itatinga soils than in Kissoko soils (Figure 1C), with values increasing 2-fold in Ac and Euc soils and 3.2-fold in Nat soils. At Itatinga, respiration rates increased significantly in the order Nat > Euc > Ac. At Kissoko, no significant difference was observed based on soil vegetation.

### 2.2. Eucalyptus Growth

When eucalypt seedlings were grown in soil samples, the original vegetation and fertilization providing nonlimiting amounts of N induced contrasting effects on total plant biomass at each site (Figure 2A,B). At Itatinga, total plant dry weight was not significantly influenced by the original vegetation of the soil samples (Euc, Ac, or Nat), with or without fertilization. In contrast, total plant biomass was increased by a factor of four in Euc soil, three in Ac soil, and two in Nat soil compared with unfertilized plants (Figure 2A). The same trends were observed for individual organs, with no significant effect of the original vegetation and a highly significant effect of fertilization (Table 1). The largest fertilization effect was observed on the stem biomass of plants grown in Ac and Euc soils, with a 5.7-fold increase compared with unfertilized conditions. Leaf biomass was also greatly increased by fertilization (×4 and ×3.5 in Ac and Euc soils, respectively). Plants grown in Nat soil had the same leaf biomass regardless of the fertilization regime. In the roots, fertilization had a more modest effect, increasing values by a factor of two, on average, compared with unfertilized conditions.

In contrast to Itatinga, the original vegetation had a significantly high effect on the total biomass of eucalypt plants grown in soil samples from Kissoko (Figure 2B). The lowest biomass was observed in plants grown in Nat soil, and those measured for Euc and Ac soils were 1.7 and 3.4 times higher than in Nat soil, respectively. Fertilization providing N supply at high concentrations significantly increased plant growth. Increment factors ranged from 1.3 in Ac soil up to 3.2 in Nat soil, with the highest biomass measured in plants grown in Ac soil (Figure 2B). The biomass of individual organs was also significantly modified by the original vegetation and fertilization (Table 2). Biomass of roots, stems and leaves of unfertilized plants varied in the order of original vegetation: Ac > Euc > Nat. With fertilization, root and leaf biomass of plants grown in Ac soil was always higher than that in Euc and Nat soils. Conversely, stem biomass of plants grown in Ac and Euc soils was similar and significantly higher than stem biomass in Nat soil.

### 2.3. N Accumulation in Eucalypt Plants

The effects of the original vegetation and fertilization rates on total plant N accumulation were the same as those observed for total biomass (Figure 2C,D). At Itatinga, without fertilization, the highest amounts of N were measured in eucalypt seedlings grown in Nat soil and the lowest amounts were measured in Euc soil, whereas plants grown in Ac soil constituted an intermediate between these extreme values (Figure 2C). As expected, chemical fertilization strongly increased N accumulation in the plants by a factor of about 4.5 in Ac and Nat soils, and by a factor of 7.7 in Euc soils. N concentrations were affected by fertilization only in roots and leaves, and by both treatments (Veg and Fert) in stems (Table 1). Without fertilization, N concentrations varied in the order leaves > roots > stem, regardless of the original vegetation. Fertilization significantly increased N concentrations in all organs of plants grown in Nat soil, but only in the leaves and stems of plants grown in Euc soil. However, fertilization did not significantly alter N concentrations measured in individual organs from plants grown in Ac soil.

At Kissoko, the amounts of total N measured in eucalypt seedlings grown without fertilization varied in the same order as total biomass depending on the original vegetation: Ac > Euc > Nat (Figure 2D). Chemical fertilization significantly increased total N accumulation by a factor of 3.4 (Ac soil), 4.5 (Euc soil), and 7.2 (Nat soil). However, for both fertilization regimes, the highest amount of N was always measured in plants grown in the Ac soil. N concentrations were only affected by fertilization only in stems and leaves, and by both treatments (Veg and Fert) in roots (Table 2). Without fertilization, N concentrations varied in the same order as at Itatinga (leaves > roots > stem), regardless of the original vegetation. Fertilization significantly increased N concentrations in all organs, regardless of the original vegetation, except in leaves from plants grown in Nat soil.

## 3. Discussion

The introduction of N_2_-fixing trees (NFT) in forest plantations is intended to improve wood production by supplying N from biological N_2_ fixation [41,42]. This practice is an attractive option in tropical forest plantations where N fertilizers are becoming expensive for foresters [41], as well as for promoting nature-based solutions in silviculture. N-rich litter increases decomposition rates by reducing the C:N ratio, as has been reported in litter from N_2_-fixing trees and mixed plantations [23,43,44,45,46]. In our study, we hypothesized that planting *Acacia* trees after several decades of eucalypt cultivation can increase soil N bioavailability for eucalypt trees. We assessed the effect of *Acacia* on the N nutrition of a non-N_2_-fixing species at two tropical sites by quantifying the N-mineralization rates of different soils under controlled conditions, as well as the actual N bioavailability to plants using a bioassay.

### 3.1. Effect of Acacia Trees on N-Mineralization Rates

At both sites, N mineralization rates ranged from 0.18 to 0.4 mg mineralized N per kg of soil per day. Measured under laboratory conditions, the average N mineralization rates in soil samples collected worldwide and from different biomes ranged from 0.2 to 0.4 mg N mineralized per kg of soil per day, with no significant differences between vegetation type divided into four categories: coniferous trees, deciduous and deciduous broadleaf trees, grasses, and shrubs [47]. Thus, although the temperature during incubation was lower than that used in our study (20 °C in [47] and 28 °C in our conditions), the rates measured in our soil samples were within the range of the values reported by Colman and Schimmel [47] for deciduous/broadleaf trees. Net N-mineralization rates were also measured in situ at both sites in pure *Acacia* and *Eucalyptus* plantations [29,32]. At both sites, the average net N mineralization in *Acacia* soil was about twice as high as in eucalypt soils. Moreover, while the laboratory-measured N-mineralization rates are in good agreement for the Congo site, those for the Brazilian soils gave nonsignificantly different average N-mineralization rates in the two soil types. This discrepancy between the two methods is difficult to explain. However, it could be due to the use of only mineral soil in the laboratory measurements. In contrast to Brazil, soil collected in *Acacia* plots in the Congo displayed higher N-mineralization rates than soil collected from eucalypt and native vegetation plots. This could be due to the high rate of N_2_ fixation by *A. mangium* measured at this site, which was estimated at 276 kg ha^−1^ [48]. Remarkably, N mineralization in soil collected under native vegetation at Kissoko was dominated by NH_4_^+^ production, in agreement with in situ N-mineralization measurements [21]. This could be due to the inhibition of nitrification induced by perennial grass species present in these savannas, as already reported by different authors [49,50,51,52]. Conversely, nitrate was the main form of mineralized N in soils sampled from *Acacia* plots at both sites, suggesting that *Acacia* trees are capable of shifting microbial populations toward nitrifying microbes. This hypothesis is supported by the increase in the ammonia-oxidizing archaea (AOA) communities involved in nitrification in *Acacia* soils compared with eucalypt soils in the Congo (Robin et al., unpublished) and at our Brazilian site [53].

### 3.2. Effect of Acacia Trees on N Bioavailability for Eucalyptus Cultivation

Considering N-mineralization rates, one would expect higher N bioavailability in *Acacia* soils at the Brazilian site than the Congolese site. However, this is not what we observed, as eucalypt seedlings cultivated in *Acacia* soils at Itatinga accumulated half as much N as at Kissoko. This lower N bioavailability at Itatinga could be due to competition between N immobilization in the microbial biomass promoted by root exudates and root uptake, which is in line with results previously reported by Waithaisong et al. [34]. This hypothesis is supported by soil respiration rates that are twice as high at Itatinga as at Kissoko in Acacia soil. Furthermore, we cannot exclude an interaction between N and P because P bioavailability was much lower at Itatinga than at Kissoko [34]. Therefore, our results demonstrate that measuring N-mineralization rates alone in the laboratory is not sufficient to predict the actual bioavailability of N to a plant, which may depend primarily upon microbial physiology determining when N will be mineralized or immobilized in the microbial biomass [54]. Compared with the native vegetation soil, the positive effect of the *Acacia* soil on growth and mineral nutrition of eucalypts was very strong at Kissoko. These results therefore show the value of using NFT species capable of fixing large amounts of N, not only in mixed-species plantations but also between two non-NFT rotations, the latter system being easier to manage for foresters.

The effects of N fertilization on eucalypt growth were very different among sites. In Brazil, we measured a very large increase in plant biomass in response to N fertilization regardless of previous vegetation, indicating that soil nitrogen is limited in all three soil types: after an *Acacia* or eucalypt crop, as well as in soil collected under native vegetation. In contrast, in the Congo, N fertilization only moderately increased the biomass of eucalypts grown in *Acacia* soil. This may indicate that under these soil conditions characterized by high P availability [34,35], N input originating from N_2_ fixation is almost equivalent to that provided by N fertilization. In contrast, N fertilization increased the allocation of biomass and N to aboveground plant parts, regardless of soil and site. In particular, the biomass of the stem was markedly increased (×5.8) when eucalypts were grown in Ac or Euc soils at Itatinga, highlighting the importance of managing soil N availability for timber production.

## 4. Materials and Methods

### 4.1. Site Description

We used two sites previously described in Waithaisong et al. [35]: one located in Brazil, in São Paulo state (Itatinga site), and the other in the Congo, on the Atlantic coast of Pointe-Noire (Kissoko site). Annual rainfall was comparable at the two sites (1370 mm at Itatinga and 1430 mm at Kissoko) with a mean annual temperature of 20 °C at Itatinga and 25 °C at Kissoko. Both sites were first afforested with *Eucalyptus* in 1940 at Itatinga, with *E. grandis* W. Hill ex Maiden, and since 1984 at Kissoko, with a hybrid between *E. grandis* and *E. urophylla* S.T. Blake (*E. urophylla × grandis).* Subsequently, plots were planted with *Acacia mangium* (Willd.) in May 2003 at Itatinga and in May 2004 at Kissoko, to quantify the effects of different silvicultural practices on stemwood production [7]. The native ecosystems prior to afforestation were tropical savannas. In Itatinga, the native savanna (called Cerrado) is an open grassland with a wide range of native woody Angiosperm species [55] with trees 3–5 m tall providing 15–40% cover [56]. The herbaceous layer was dominated by species belonging to the family *Poaceae* [56]. At Kissoko, the savanna was dominated by grass species of the family *Poaceae*, which were *Loudetia arundinacea* (Hochst.) [57], *L. simplex*, and *Hyparrhenia diplandra* [58].

The soils were ferralsols at Itatinga and ferralic arenosols at Kissoko [59]. Both soils were acidic and sandy, with low content of exchangeable elements and a low cation exchange capacity (1.76 and 0.82 cmolc.kg^−1^ in plantation soils at Itatinga and Kissoko, respectively). Despite the total P content being the same order of magnitude at both sites (0.21 and 0.28 g P kg^−1^ at Itatinga and Kissoko, respectively), the soil was dominated by organic P (Po) at Itatinga and by mineral P (Pi) at Kissoko [34,35]. Total N content was higher at Itatinga (0.7 g kg^−1^ dry soil) than at Kissoko (0.54 g kg^−1^ dry soil).

We used field trials set up in May 2003 at Itatinga [27,28] and in May 2004 at Kissoko [54], consisting of pure stands of eucalypts or acacias, planted at densities of 1111 and 800 trees per hectare at Itatinga and Kissoko, respectively. At planting, starter fertilization was applied to eucalypt and *Acacia* trees within a radius of 50 cm around each tree [7]. Starter fertilization varied by site, with P (40 kg ha^−1^ as superphosphate), K (75 kg ha^−1^ as KCl) and N (120 kg ha^−1^ as ammonium nitrate) applied at Itatinga [7], and only N as ammonium nitrate (43 kg ha^−1^) at Kissoko [57]. The N_2_ fixation in *Acacia* plots was higher at Kissoko than at Itatinga [32,33,60].

### 4.2. Soil Sampling

At each site, there were three treatments, consisting of monospecific *Acacia* (Ac), monospecific eucalypt (Euc), and nearby native vegetation on the same soil type (Nat), with three blocks for each treatment. The soils used for N-mineralization and respiration measurements were mineral topsoil (0–10 cm, without organic layer nor leaves) and of the same provenance as those used in Waithaisong et al. [34,35]. They were collected at the end of the rainy season at both sites. At Itatinga, the soil was collected in February 2012 at the end of two rotations lasting 8 years each. At Kissoko, the soil was collected in May 2011 at the end of one rotation lasting 7 years. Soils were air-dried, sieved to 2 mm and stored at room temperature pending analysis.

Soils for the pot experiment were collected in November 2013 at both sites. Cores (5 cm in diameter, 20 cm in height, and corresponding to approximately 300 g of soil) were used to collect the topsoil from plots of pure *Acacia*, eucalypt, or native vegetation identical to those used for respiration and N-mineralization measurements. In the planted forest plots (Euc or Ac), three soil cores were taken around one tree, whereas in the native vegetation plots, the soil cores were taken along a 10 m transect. After the removal of the organic horizon and leaves, a composite sample was then formed from all soil cores from a given treatment before filling each pot with 300 g of wet soil.

### 4.3. Measurement of Soil Respiration

Basal soil C respiration was measured using the procedure described by Hamdi et al. [61]. Subsamples of 25 g of most soils were placed in sealed jars after adjusting their water potential (pF) to 2.5. The jars (n = 9 per soil) were incubated twice at 28 °C for 28 days. The first period was for preincubation and the second was for respiration measurements. Soils were incubated with an alkaline trap (15 mL 0.5 M NaOH). Traps were changed at 3, 7, 14 and 28 days and analyzed within the day. The rate of basal respiration was estimated by titrating unreacted NaOH with 1 M HCl to determine the CO_2_ released.

### 4.4. Bioassay

Bioassay experiments were carried out in Brazil at the nursery of the Itatinga Station nursery and in the Congo at the Experimental Station of Pointe-Noire (CRDPI). Seedlings were obtained from seeds of *E. grandis* at Itatinga (supplied by the company Suzano, Brazil “https://www.suzano.com.br/en/ (accessed on 7 February 2023)” and from *E. urophylla × grandis* cuttings (clone 18-147) at Kissoko. One 1–2 cm-tall seedling was transferred to a pot containing the topsoil collected from the Euc, Ac or Nat plots to start the experiment. Two fertilization regimes were applied to the seedlings by irrigation. The first consisted of demineralized water alone (no fertilization, −Fert). The second (fertilization, +Fert) followed the NK fertilization regime classically used by foresters to manage eucalypt plantations [58] with mineral N (4 mM), combined with K (5 mM) on both sites. For the N source, we used (NH_4_)_2_SO_4_ at Itatinga and NH_4_NO_3_ at Kissoko to follow each country’s commercial nursery protocols, as well as using KCl at both sites. K was applied in addition to N because soil K availability can be low in tropical soils and K deficiency greatly reduces the growth of unfertilized eucalypt trees at Itatinga [62,63]. Even though the main reason for adding fertilizer in our study was to assess how N addition (through biological fixation or fertilization) affected eucalypt growth, we decided to include K in the fertilization treatment to avoid any confounding effects limiting the growth and nutrient uptake of eucalypt seedlings resulting from K deficiency. The number of plants per soil origin and treatment (+Fert or −Fert) was 6 at Itatinga and 10 at Kissoko. Plants were harvested 7 (Itatinga) and 6 (Kissoko) months after planting. They were separated into roots, stems and leaves before drying (60 °C) and weighing. The plant material was stored in plastic bags at room temperature pending N measurements carried out in France.

### 4.5. Chemical Assays

To measure mineralized N during the C respiration experiment, we used soil to fill the jars, and after 28 days of incubation we extracted mineral N with 1 M KCl (1/10, *w*/*v*). Mineral N (ammonium and nitrate) was then assayed with a continuous flow analyzer (CFA, SKALAR) “https://fr.skalar.com (accessed on 7 February 2023)”. The total N content of the roots, stems, and leaves was measured in finely ground material and redried overnight at 50 °C. Mineralization was carried out on approximately 10 mg of dry material using acid digestion with 36 N H_2_SO_4_ and salicylic acid (5%, *w*/*v*), as described in [34]. After heating at 330 °C and inducing the volatilization of organic C with ultrapure H_2_O_2_ (110 volumes, not stabilized with phosphate), the transparent solution of H_2_SO_4_ (36 N) was diluted to 0.1 N with ultrapure water and ammonium was quantified with a continuous flow analyzer, as above.

### 4.6. Data Analysis

All data analyses were performed with R stat (R 3.6.2 version). The effect of treatments on N-mineralization and soil respiration rates were analyzed with a one-way ANOVA and plant biomass and N accumulation were analyzed with a two-way ANOVA. The homogeneity of variances and the normality of the ANOVA residuals were verified using the Levene and Shapiro–White tests, respectively. If these tests were significant (*p* < 0.05), the data were log-transformed before analysis. The differences between means were analyzed using Tukey’s HSD post hoc test and the “predictmeans” package (version 1.0.4).

## 5. Conclusions

Our study shows that bioassays conducted by growing a non-N_2_-fixing target tree species on soils sampled under N_2_-fixing trees can be an effective tool for evaluating the potential benefit of rotations alternating N_2_-fixing and non-N_2_-fixing stands. Bioassays take into account the complex interactions between roots and soil biogeochemical properties, allowing for quantification of actual soil N bioavailability for non-N_2_-fixing tree species in a more reliable manner than simple measurements of soil N mineralization. We also show that the positive effect of N_2_-fixing tree species on N bioavailability for non-N_2_-fixing trees is site-dependent. This work highlights the great complexity of biogeochemical interactions and the need for multisite studies to determine the effects of introducing N_2_-fixing species for forest management. The bioassay technique could be used to establish a first screening on many sites of the effects of the introduction of N_2_-fixing species on the growth and mineral nutrition of non-nitrogen-fixing species grown in successive rotations.

## Figures and Tables

**Figure 1 plants-12-00802-f001:**
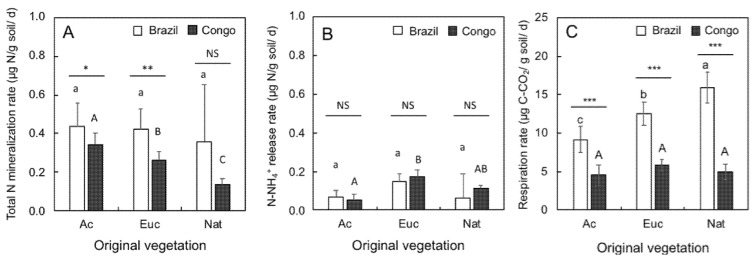
Rates of total nitrogen mineralization (N-NH_4_^+^ + N-NO_3_^−^) (**A**), N-NH_4_^+^ production (**B**), and respiration (**C**) measured in soil samples taken from topsoil (0–20 cm) under different vegetation of origin: *Acacia* (Ac), *Eucalyptus* (Euc), and native vegetation (Nat) in Brazil (Itatinga, white bars) and Congo (Kissoko, dark grey bars). Each bar represents mean (n = 9) with a confidence interval (*p* = 0.05). Different letters on bars (lowercase for Brazil and uppercase for Congo) indicate significant differences among original vegetation (one-way ANOVA and Tukey’s HSD post hoc test at *p* ≤ 0.05). Asterisks indicate site effects (Student’s *t*-test): NS (*p* > 0.05), * (*p* < 0.05), ** (*p* < 0.01), *** (*p* < 0.001).

**Figure 2 plants-12-00802-f002:**
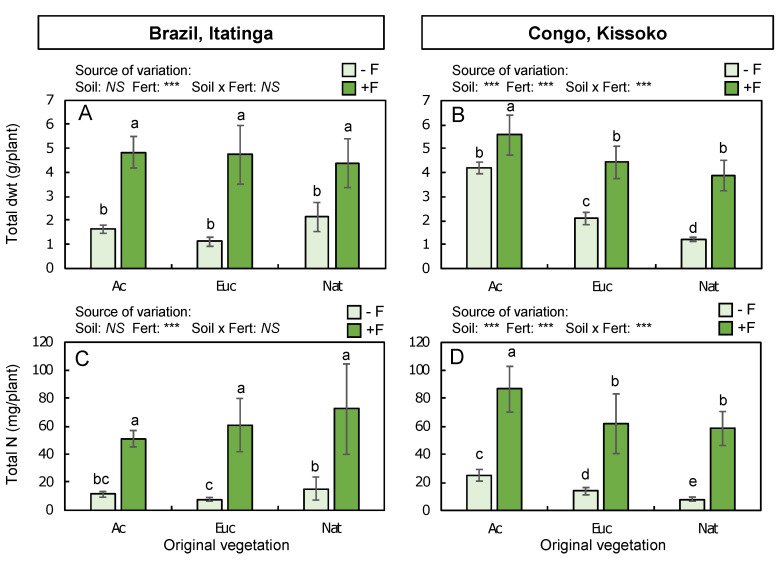
Total dry weight (dwt) (**A**,**B**) and N (**C**,**D**) accumulation in 6-month-old eucalypt seedlings grown in pots containing soils collected from the topsoil (0–20 cm) under different original vegetation (Veg), which were *Acacia* (Ac), *Eucalyptus* (Euc), and native vegetation (Nat) in Brazil (Itatinga) (**A**,**C**) and in the Congo (Kissoko) (**B**,**D**). Plants received no fertilization (−Fert, light green) or chemical N fertilization (+Fert, intense green) that was provided (nonlimiting) in the irrigation solution. Each bar represents the mean (n = 6 in Brazil, n = 10 in the Congo) with standard deviation. Results of plant biomass without fertilization are extracted from [34]. For each variable within a site, effects of original vegetation (Veg), fertilization (Fert), and their interaction (Veg x Fert) were analyzed with a two-way ANOVA: NS (*p* > 0.05), *** (*p* < 0.001). Within each site, different letters indicate significant differences in means based on original vegetation and fertilization (comparison of means and Tukey’s HSD post hoc test, *p* < 0.05).

**Table 1 plants-12-00802-t001:** Dry weight (dwt) and N concentrations (N conc.) measured in roots, stems, and leaves of 7-month-old eucalypt grown in pots containing soils collected from 0–20 cm under different original vegetation (Veg), which were *Acacia* (Ac), *Eucalyptus* (Euc), and native vegetation (Nat) in Brazil (Itatinga). Plants received no fertilization (−Fert) or fertilization (+Fert) consisting of N ((NH_4_)_2_SO_4_) and K (KCl) supplied in irrigation solution with unrestricted availability. Results of plant biomass without fertilization are extracted from [34]. Each value is the mean (n = 6) with standard deviation between brackets. For each organ and parameter, the effects of original vegetation (Veg), fertilization (Fert) and their interaction (Veg × Fert) were analyzed with a two-way ANOVA, with the following levels of significance: NS (*p* > 0.05), * (*p* < 0.05), *** (*p* < 0.001). Means were compared using a Tukey’s HSD post hoc test and different lowercase letters accompanying the means indicate significant differences at *p* < 0.05.

		Fertilization and Original Vegetation								
		−Fert			+Fert			Two-Way ANOVA		
Variable	Organ	Ac	Euc	Nat	Ac	Euc	Nat	Veg	Fert	Veg × Fert
dwt (g/plant)	Roots	1.05 b (0.16)	0.56 b (0.14)	1.14 b (0.34)	2.17 a (0.32)	2.60 a (0.87)	2.57 a (0.79)	NS	***	NS
	Stem	0.18 b (0.04)	0.16 b (0.05)	0.35 b (0.20)	1.04 a (0.27)	0.93 a (0.10)	0.87 a (0.35)	NS	***	NS
	Leaves	0.41 c (0.03)	0.39 c (0.11)	0.66 bc (0.24)	1.63 a (0.33)	1.36 a (0.38)	1.16 ab (0.47)	NS	*****	*
N conc. (mg/g dwt)	Roots	6.25 b (1.60)	6.69 b (2.07)	6.67 b (1.84)	9.37 ab (0.78)	8.79 ab (1.30)	12.70 a (5.04)	NS	***	NS
	Stem	4.42 cd (0.69)	4.51 cd (0.52)	3.43 d (1.42)	8.40 bc (2.31)	11.38 b (3.21)	17.61 a (8.78)	***	***	***
	Leaves	9.83 b (2.80)	8.56 b (2.19)	9.60 b (4.20)	14.15 ab (4.86)	22.21 a (4.37)	22.60 a (7.40)	NS	***	NS

**Table 2 plants-12-00802-t002:** Dry weight (dwt) and N concentrations (N conc.) measured in roots, stems, and leaves of 6-month-old eucalypt grown in pots containing soils collected from 0 to 20 cm under different original vegetation (Veg), which were *Acacia* (Ac), *Eucalyptus* (Euc), and native vegetation (Nat) in the Congo (Kissoko). Plants received no fertilization (−Fert) or fertilization (+Fert) consisting of N (NH_4_NO_3_) and K (KCl) supplied in irrigation solution with unrestricted availability. Results of plant biomass without fertilization are extracted from [34]. Each value is the mean (n = 10) with standard deviation between brackets. For each organ and parameter, effects of original vegetation (Veg), fertilization (Fert), and their interaction (Veg × Fert) were analyzed with a two-way ANOVA, with the following levels of significance: NS (*p* > 0.05), * (*p* < 0.05), ** (*p* < 0.01), *** (*p* < 0.001). Means were compared using a Tukey’s HSD post hoc test and different lowercase letters accompanying the means indicate significant differences at *p* < 0.05.

		Fertilization and Original Vegetation								
		−Fert			+Fert			Two-Way ANOVA		
Variable	Organ	Ac	Euc	Nat	Ac	Euc	Nat	Veg	Fert	Veg × Fert
dwt (g/plant)	Roots	0.73 ab (0.12)	0.45 c (0.08)	0.27 d (0.06)	0.79 a (0.13)	0.60 bc (0.14)	0.57 bc (0.25)	***	*****	*
	Stem	1.2 b (0.13)	0.54 c (0.06)	0.31 d (0.04)	1.77 a (0.26)	1.54 a (0.20)	1.15 b (0.29)	***	***	**
	Leaves	2.27 b (0.16)	1.10 c (0.16)	0.65 d (0.05)	3.00 a (0.48)	2.28 b (0.60)	2.15 b (0.37)	***	***	**
N conc. (mg/g dwt)	Roots	6.82 c (1.09)	7.48 c (1.75)	6.92 c (1.96)	13.71 a (1.68)	12.97 a (2.58)	10.03 b (2.66)	*	***	**
	Stem	2.44 c (0.68)	2.76 c (0.98)	3.03 bc (1.07)	4.90 ab (2.76)	5.09 ab (1.42)	5.86 a (2.20)	NS	***	NS
	Leaves	9.83 b (1.54)	9.76 b (1.56)	9.60 b (1.32)	22.60 a (3.75)	22.21 a (4.60)	14.14 ab (3.53)	NS	***	NS

## Data Availability

The data presented in this study are available on request from the corresponding author.

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
