# Peer review of "Potential of Bioassays to Assess Consequences of Cultivation of Acacia mangium Trees on Nitrogen Bioavailability to Eucalyptus Trees: Two Case-Studies in Contrasting Tropical Soils"

_plants, 2023, doi:10.3390/plants12040802_

Round 1

Reviewer 1 Report

The topic is original and relevant. The study focuses the effects of different types of vegetation associated with eucalyptus plantation and N fertilization in two locations (Brazil and Congo). The limitation of the study  is the short-term experiment. The effect of climate is not evaluated.

The manuscript requires an improvement on the presentation, including a higher clarity, e.g. using shorter sentences. Language also requires a revision. Overall, keywords, introduction and M&M are well addressed, but more clarity is required, especially the experiment used.

Minor issues: 

-L46: what about the environmental pollution?

-L100: what about the "Nat"?

-L100/102: unclear. Please improve

-L110: write "nitrification was not significant in Nat..."

-L131: write: "total plant biomass"

-L164; write: "significantly high effect"

-L166: delete "amounts" and write "was observed"

-L168: "Non-limiting N fertilization": what do you mean?

-L172: please verify the sequence order

-L191: verify if the highest value is at Euc soil

-L195: write "(Table 1)"

-L332: delete "$"

Fig. 1: Legend should indicate the colours used (Fig. 1A). It is unclear which one represents NH4+ release. Also, clarify the letters used for means separation (Idem for Fig. 2).

-L332: 

Reviewer 2 Report

Dear Authors,

I have reviewed the paper "  Potential of bioassays to assess the consequences of the cultivation of Acacia mangium trees on nitrogen bioavailability to Eucalyptus trees. Two case-studies in contrasting tropical soils   ". The aims of the paper are germane with Plants topic. The paper is written with a moderate English level. The contribution of this paper to the scientific knowledge is moderate. In my opinion there some flaws and I suggest the corrections in the comments for the authors  in the file attached. 

Round 2

Reviewer 1 Report

Authors have improved the manuscript but still several issues are found, as follows:

Language requires a major improvement.

Conclusions are overall not well addressed. Conclusions seem more an Abstract. There is a need to make a strong improvement.

Specific comments:

-Actual L54/55: please delete the sentence "Conversely...However,"

-L56: write "0.242 kg) N-N2O ha-1"

Please demonstrate (for instance, using a reference) that Acacia and other fixing legumes emitt N2O to the atmosphere

-L60: Describe how did you manage the Acacia and Eucalyptus.

-L71/74: improve the sentence, and clearly explain why there is a less N requirement after 3-years of eucalyptus plantantion. Why is soil N mineralization important only at this phase?

-L181/184: improve the comment on statistic results.

-L434, L440: delete "samples"

-L442: how long is the rotation? I cannot understand this sentence.

-L443/446: I cannot understand this sentence.

-L447: did authors evaluate "geo" cycling?

Fig. 1: improve the legend.
